# Phylogenetic Analysis of the Complete Mitochondrial Genomes in the Ten *Rupicapra* Subspecies and Implications for the Existence of Multiple Glacial Refugia in Europe

**DOI:** 10.3390/ani12111430

**Published:** 2022-06-01

**Authors:** Trinidad Pérez, Margarita Fernández, Borja Palacios, Ana Domínguez

**Affiliations:** 1Departamento de Biología Funcional (Genética), Facultad de Medicina 6ª Planta, Universidad de Oviedo, C/Julián Clavería 6, 33006 Oviedo, Spain; margarifr@hotmail.com (M.F.); sanjurjo@uniovi.es (A.D.); 2Parque Nacional Picos de Europa, Avda. Covadonga 43, 33550 Cangas de Onís, Asturias, Spain; bpalacios@pnpeu.es

**Keywords:** chamois, eurasia, glacial refuges, intraspecific variation, mitogenome, phylogeny, pleistocene

## Abstract

**Simple Summary:**

The successive glaciations that took place during the Pleistocene shaped de distribution of temperate species in Europe, both plants and animals. Traditionally, it has been hypothesized that during the coldest periods, species took refuge in three areas of southern Europe (Iberian, Italian and Balkan peninsulas) and then recolonized the north when temperatures rose and conditions were more favorable. In the present work, the complete mitochondrial sequences of the ten described chamois populations have been analyzed to identify which areas of the continent have served as refuges for the species during the glaciations. The results of the present work are consistent with the existence of multiple glacial refugia across Europe, revealing a much more complex picture of the effect of glaciations on the genetics of temperate species than is commonly accepted, giving us a better understanding of how past events determine the present species.

**Abstract:**

The current distribution of populations in Europe is marked by the effects of glaciations that occurred during the Pleistocene. Temperate species were isolated in glacial refugia that were the sources of postglacial recolonization. The traditional glacial refuge areas were the Iberian, the Italian and the Balkan peninsulas. Here we revisit the evolutionary history of chamois (*Rupicapra* genus) to evaluate other sites in continental Europe and Anatolia that have been suggested as potential refuges. We have obtained the complete mitochondrial sequence of seven chamois, including the subspecies *parva*, *carpatica*, *caucasica*, and *asiatica* whose mitochondrial genome had not been yet reported. These, together with the other fourteen sequences already in the GenBank, represent the different geographical populations of the *Rupicapra* genus. The phylogenetic analysis showed the three old clades, dating from the early Pleistocene, already reported: mtW in the Iberian Peninsula, mtC in the Appenines and the Massif of Chartreuse, and mtE comprising all the population from the Alps to the east. The genomes within each of the clades mtW and mtE, showed divergence times larger than 300 thousand years. From here, it can be argued that the present-day lineages across Europe are very old and their split dates back to the middle Pleistocene.

## 1. Introduction

The effect of Pleistocene glaciations in shaping the genetic structure of temperate species in Europe has been the object of many studies since the pioneer phylogeographic studies of Taberlet et al. [1] and Hewitt [2]. Variations in climatic conditions lead to the restriction of animal species to glacial refugia during glacial maxima and the posterior recolonization when climatic conditions were favorable. The pattern revealed for many plant and animal species led to the widely accepted hypothesis of three southern refugia in the southern peninsulas (Iberia, Italy, and the Balkans) and the posterior recolonization of the northern wards. However, the consideration of faunal assemblages of archeological sites during the Last Glacial Maximum (23 to 16 kya) showed the presence of temperate species in southwestern France and the Carpathian regions [3,4,5]. In addition, some phylogeographic studies suggested that areas of the south of the Caucasus constituted a further refugium to the East [6].

Phylogeographic studies on the chamois, genus *Rupicapra*, based on mtDNA revealed three very old clades [7,8] with a clear geographical signal in its distribution across Eurasia. The comparison of the complete mitochondrial genomes of these three clades with the genomes of Caprini highlighted the ancient presence of chamois in Europe that was estimated to be around 1.9 million years ago (mya) [8,9,10] in the Early Pleistocene. This is by far older than the age of the most ancient *Rupicapra* fossils in Europe that were found in the Balkans and corresponded to the beginning of the middle Pleistocene, between 780 and 750 kya [11]. The distribution pattern of the three old mitochondrial clades shows that there were no large migrations of female chamois during the glaciations of the Pleistocene.

At present, the genus *Rupicapra* inhabits most mountain regions in Eurasia, from the Cantabrian Mountains in the southwest of Anatolia and the Caucasus. The populations occupying the different mountain ranges were given the subspecies category and the most accepted taxonomy [12] groups placing these ten subspecies into two species (see Figure 1), although we have argued that this classification is not supported by genetic data [13,14]. The three old mitochondrial clades were referred to as west, central, and east (mtW, mtC, and mtE) according to its distribution [8]. The clade mtW is almost limited to the Iberian Peninsula, but it was also present at low frequency in the west Alps. The clade mtC is shared by the population *ornata* in the Apennines and *cartusiana* in the Massif of Chartreuse, while the clade mtE is present in all the populations from the Alps to the east. This distribution was attributed to the ancient division of matrilineal clades isolated by the glacial sheets of the Alps and the Pyrenees during glacial periods and limited recolonization of female herds during periods of expansion/contraction of populations during glacial/interglacial cycles in the central part of the distribution. It is noteworthy that Camerano had identified these three major lineages (which he accepted as valid species) on the basis of morphometry [15]. The study of nuclear markers [8,14] revealed a much younger divergence between populations than that observed from mtDNA, and the male-mediated introgression in the central part of the distribution (see Figure 1). We have attributed the enormous difference in divergence time estimates that are obtained when using either the mitochondrial or the nuclear sequences to the effects of strong female philopatry together with rampant male dispersal at the end of the Pleistocene.

As said, the three main mitochondrial clades are very old, the posterior differentiation within each of these clades must have occurred during the glaciations and its study would allow making inferences about minor glacial refugia and recolonization. The objective of the present study is to determine the divergence time of mitochondrial subclades within the clades mtW and mtE through the analysis of complete mitochondrial genomes of individuals representing the different populations and evaluate the possibility that continental areas had served as glacial refugia.

## 2. Materials and Methods

In this study, we obtained the mitochondrial genome of seven chamois to cover the different geographical populations and compared them with the sequences already available in the GenBank. The DNA of the samples had been previously isolated [17,18] and the sequencing was done using the Sanger sequencing method with the set of 23 primer pairs published by Hassanin et al. [19] and following the protocol already described [10]. We have included two individuals of *R. r. balcanica* after the observation, in a previous work based on the mitochondrial CR [20], of the large diversity of mitochondrial sequences in this subspecies. Unfortunately, individuals from Greece representing other lineages closer to *caucasica* and *asiatica* could not be included due to the low quality of the available DNAs.

The seven mitochondrial genomes of *Rupicapra* newly obtained, together with the fourteen genomes previously present in the GenBank, make a total of 21 sequences among which are represented the different subspecies of chamois. We investigate the phylogenetic relationships of these sequences using the sequence of *Capra ibex* (FJ207526.1) as an outgroup. The repeats at the control region were excluded from the analysis due to ambiguities in the alignment so that the final number of sites was 16,272. The phylogeny was studied by Bayesian analysis using the Monte Carlo Markov chains (MCMC) method implemented in BEAST 2.1 [21]. We used a Relaxed Clock Log-Normal Model and a Coalescent Constant Population as prior. The model of nucleotide substitution was the GTR + G+I with the empirical base frequencies, as determined by the AIC criteria in MEGA 7 [22]. Divergence times were estimated with BEAST 2.1, using as calibration points the divergence of Caprini [23] with a normal prior and mean age = 8.9 Ma (standard deviation = 2 Ma) and the *Rupicapra* node [10] with an age of 1.93 Ma (standard deviation = 0.43 Ma). All the analyses were run for 25 million generations with tree and parameter sampling every 1000 generations. A burn-in of 10% was used, and the convergence of all parameters was assessed using the software Tracer 1.7 [24]. The maximum clade credibility criterion tree was obtained with TreeAnnotator, using a burn-in of 10% and with mean node heights. The reliability of the nodes was assessed by the posterior probability (BPP) of the nodes under the Bayesian approach. The topology of the tree was further investigated by other three methods of tree reconstruction using the software MEGA7. We made a Neighbor-Joining (NJ) tree based on Jukes–Cantor distance under the complete deletion option. A model-free Maximum Parsimony (MP) tree was obtained using the Close-Neighbor-Interchange algorithm with search level 3, in which the initial trees were obtained with the random addition of sequences (10 replicates). The MP consensus tree was inferred from 1000 bootstrap replicates with MEGA. A Maximum Likelihood (ML) tree was obtained with the Heuristic Method of the Nearest-Neighbor Interchange. The reliability of the nodes under NJ, MP and ML was assessed by 1000 bootstrap replicates. The trees obtained by the different methods were visualized with FigTree 1.4.2 [25].

## 3. Results and Discussion

The complete mitochondrial genomes of the seven newly sequenced samples were deposited in the GenBank and their accession numbers, as well as the other sequences used in this work, are given in Table 1 and Table 2 respectively. The total lengths of the sequences were between 16,394 nt for *carpatica* and 16,438 nt for *parva*. The mitochondrial genomes presented the standard composition of 13 protein-coding genes, 22 tRNAs, 2 rRNAs, the replication origin of the light strand, and the CR. The overall nucleotide composition of the H strand was 33% A, 27% C, 14% G, and 26% T, similar to the mtDNA of the other *Rupicapra* mitochondrial genomes. The CR of the newly sequenced *Rupicapra* genomes is organized into three main domains, as is general in Caprini, with two repetitive sequences (RS) and presents a length between 838 and 849 nt, but the sequence of *R. rupicapra carpatica* displays a 41 nt deletion in the first RS relative to the other sequences. The same deletion is also present in the sequences of *R. pyrenaica ornata* and *R. rupicapra cartusiana* of the clade mtC and in some individuals of *R. pyrenaica* [8,10]. Thus, the deletion is present in the three clades and hence must have arisen independently due to the repeated characteristic of the region.

The different methods of the three reconstructions reveals the three main clades of *Rupicapra* previously described (see the Bayesian tree in Figure 2 and the NJ, MP, and ML trees in the Appendix A, respectively). The great divergence between sequences that form the clade mtW, as well as between sequences belonging to the clade mtE, can be appreciated under either method of reconstruction. The high diversity between mtDNA within those clades shows that matrilineal lineages evolved in allopatry during glacial periods. The split between *parva* and *pyrenaica* occurred 340 kya, before the major glaciations of the Pleistocene. The deep divergence between the matrilineal sequences in central Europe and the regions to the southeast of the distribution of chamois can also be noted. This indicates that, contrary to the general trend [27], there was no female-mediated recolonization from the Balkan Peninsula into the central European mountain ranges. The split of the subclades of the main clade mtE is very old, around 560 kya. Within the clade mtE, it can be noted at least three old subclades with divergence times in the order of 300 kya. There is a basal node with the sequences of asiatica, caucasica, and balcanica of North Macedonia and the High Tatras. This node is well supported by the different methods of tree construction and a similar result was attained by Rezić et al. [28] when studying the Control Region. The clustering of some Tatra chamois with the eastern-most representatives of the genus points to glacial refugia in the region of Caucasus and Anatolia and limited posterior expansion to the Tatras. The regions of southern Europe, Turkey, and the southern Caucasus had been identified as likely places for multiple glacial refugia [29] and our observation concurs with this view. Other subclades are 330 kya old, group sequences from Serbia and Croatia (some of them are supposed to be from descendants of reintroduced chamois) and the Tatra National Park (Poland side). A third subclade, dated 300 kya, groups sequences classified within the subspecies *rupicapra*, from Slovenia and Croatia and the Central Alps. Finally, the sequence from *carpatica* is basal to these three subclades, although other methods of tree reconstruction put this sequence within a group with the most central *rupicapra* samples. Thus, the present-day lineages that conform the clade mtE are old and their split dates back to the middle Pleistocene. Their differentiation must be related to periods of isolation during glacial maxima and limited recolonization of mountain regions by females during interglacial periods. The regions of Croatia south east of the Alps have been unglaciated since the Medium Pleistocene; in fact, *Rupicapra* remains from that time were identified [30]. This can be related to the old female lineages that were identified in this study, and hence, our data coincides with the alternative proposed by Miracle et al. [30] that faunal communities in the region did not dramatically change across the 40–30 kya, could have been buffered from climatic oscillations owing to microclimate and topography and that the region could constitute a glacial refugium. In addition, the survival of species in Central European refugia and the Carpathian region before the Last Glacial Maximum (23–16 kya) has been shown [3,31]. It can be noted that the region to the north of the Balkan Peninsula in the central and Eastern Europe was not glaciated during the Riss (=Saale) glacial time (MIS Marine Isotope State 6: (~200/191–140/130 kya) and that *Rupicapra* remains were identified there all along the interval of MIS 6-MIS 4, at the end of the Middle and the first half of the Late Pleistocene [5]. Our data are consistent with the idea that temperate species such as the chamois occurred across a large part of Europe in Central European refugia as well as in Anatolia and the Caucasus during the glacial periods of Pleistocene. 

## 4. Conclusions

The phylogenetic analysis of mitochondrial genomes of *Rupicapra* is consistent with the idea that temperate species such as the chamois occurred across a large part of Europe in Central European refugia during the glacial periods of Pleistocene.

## Figures and Tables

**Figure 1 animals-12-01430-f001:**
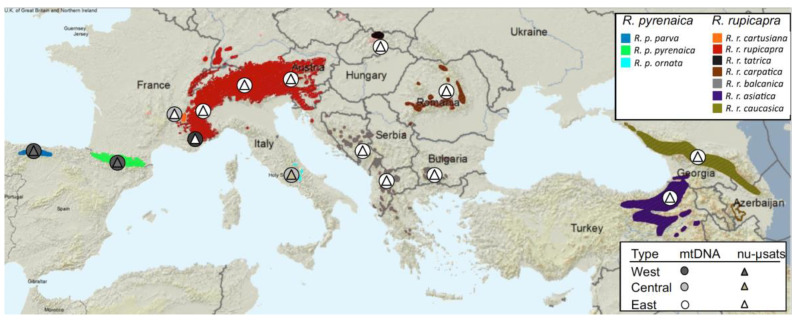
Geographic distribution of the subspecies of the genus *Rupicapra*. The map was modified from the distribution map on the IUCN Red List [16]. The affiliation to Clades West, Central, and East of extant populations of chamois for the mtDNA and nuclear markers (microsatellites and introns) is represented by forms colored in black, grey, and white, respectively.

**Figure 2 animals-12-01430-f002:**
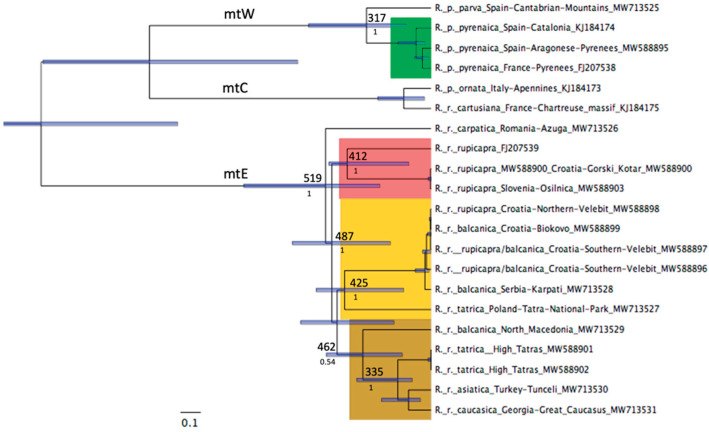
The Bayesian tree obtained with BEAST includes complete mtDNA sequences from the ten subspecies of chamois. The tree was rooted with *Capra ibex* (not shown). The Clades mtW, mtC, and mtE are indicated. The value above the subclade nodes corresponds to the mean age estimate of the nodes in thousand years, with 95% credibility intervals indicated by the blue bars. The Bayesian posterior probabilities are shown below the nodes. The colored rectangles highlight subclades within the main clades.

**Table 1 animals-12-01430-t001:** Information about the origin and GenBank accession numbers of the seven newly sequenced samples.

Subspecie	Code	GenBank Acc. Nr.	Location	Country	Year	Tissue
*R. p. parva*	CBWo24	MW713525	Leitariegos, Asturias	Spain	2006	muscle
*R. r. carpatica*	CPo06	MW713526	Azuga	Romania	2000	tooth
*R. r. tatrica*	TAo02	MW713527	Tatra National Park	Poland	1999	muscle
*R. r. balcanica*	BAo17	MW713528	Karpati	Serbia	1999	muscle
*R. r. balcanica*	BAo20	MW713529	Mavrovo National Park	North Macedonia	2013	muscle
*R. r. asiatica*	TUo01	MW713530	Tunceli	Turkey	2007	muscle
*R. r. caucasica*	CUo05	MW713531	Great Caucasus	Georgia	1996	skin

**Table 2 animals-12-01430-t002:** Information about the origin and GenBank accession numbers of the sequences retrieved from the GeneBank.

Subspecie	GenBank Acc. Nr.	Location	Country	Reference
*R. p. pyrenaica*	FJ207538	Pyrénées	France	[19]
*R. p. pyrenaica*	KJ184174	Pyrénées	France	[10]
*R. p. pyrenaica*	MW588895	Aragonese Pyrenees	Spain	[26]
*R. p. ornata*	KJ184173	Apennines	Italy	[10]
*R. r. rupicapra*	FJ207539	Cytogenetic collection 2001-175, MNHN	France	[10]
*R. r. cartusiana*	KJ184175	Chartreuse Massif	France	[10]
*R. r. balcanica*	MW588899	Biokovo	Croatia	[26]
*R. r. rupicapra*	MW588898	Northern Velebit	Croatia	[26]
*R. r. rupicapra*	MW588900	Gorski Kotar	Croatia	[26]
*R. r. rupicapra*	MW588903	Osilnica	Slovenia	[26]
*R. r. tatrica*	MW588901	National Park High Tatras	Slovakia	[26]
*R. r. tatrica*	MW588902	National Park High Tatras	Slovakia	[26]
*R. r. rupicapra*, Suspected hybrid	MW588896	Southern Velebit	Croatia	[26]
*R. r. rupicapra*, Suspected hybrid	MW588897	Southern Velebit	Croatia	[26]

## Data Availability

All sequences generated during this study have been deposited in the GenBank (https://www.ncbi.nlm.nih.gov/genbank/).

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
