# Peer review of "Phylogenetic Analysis of the Complete Mitochondrial Genomes in the Ten Rupicapra Subspecies and Implications for the Existence of Multiple Glacial Refugia in Europe"

_animals, 2022, doi:10.3390/ani12111430_

Round 1

Reviewer 1 Report

Dear authors,

interesting and nice study, congratulations! You can find my comments in the pdf document itself.

all the best

Author Response

Dear referee 1 thank you very much for your kind comments and for your review of our article. We have taken your considerations into account and made the suggested changes. Below is a detailed list of the changes made.

Best regards

I suggest to write the Iberian, Italian (or Apennine) and Balkan peninsulas

Page 1 Ln 12-13 changed

Alphabetical order???

You can delete Rupicapra because it is in the title.

Instead mitochondrial genom which is in the title write mitogenome

Page 1 Ln 34-35 changes done

This is okay, I suggest to correct the sentence in the simple summary.

Page 1 Ln 12-13 change

Italic?

Page 2 Ln 59 Yes, done

I am not sure is this correct name, the authors should check, but I think that location is the Serbian Carpathians or "Karpati"

Page 5 ln 160 table 1a changed Karpayi to Karpati

Serbia is the correct name

Page 5 ln 160 table 1a changed Servia to Serbia

Table line is missing

Page 5 ln 160 table 1a line added

Serbia is the correct name

Page 6 ln 175 changed Servia to Serbia

Poland or Slovakia? Please specify

Page 6 ln 190-191 Poland side of the Tatra National Park. Added

Here is some mistake, impossible that Hewitt (2004) quote Miaracl which was published later (2010)

Page 6 ln 201 You are right it is Miracle reference, 29 instead of 26. Changed

Serbia_Karpati

Page 7 ln 215 figure 2 Changed

Reviewer 2 Report

Perez et al. analyze complete mitochondrial genomes from the Rupicapra genus to gain insights into the distribution of the species in Europe during the Pleistocene glacial periods. Their data suggests matrilineal lineages evolved in allopatry during the glacial periods and that chamois occurred across a large part of Europe during the glacial periods of the Pleistocene.

The manuscript is well written and enjoyable to read. The methodology appears sound and I have only a few comments/suggestions that I will list below.

Ln 97. Since there are so many sequencing methods around, at the moment, could the authors indicate which sequencing method was used in this work?

Ln 140. The 41nt deletion mentioned is located in a repetitive sequence, these can be difficult to sequence/assemble depending on the method used. Can the authors be sure this is not the result of a sequencing or assembly error?

Fig. 2. What do the colored rectangles represent?

The supplement file only contained one extra tree and didn't have any legend, so I couldn't assess this file.

There are multiple instances of Gen-Bank in the text. Please correct.

Author Response

Dear referee 2 thank you very much for your kind comments and for your review of our article. We have taken your considerations into account and made the suggested changes. Below is a detailed list of the changes made.

Best regards

Ln 97. Since there are so many sequencing methods around, at the moment, could the authors indicate which sequencing method was used in this work?

Ln 106 the sequencing method was Sanger method, information has been added.

Ln 140. The 41nt deletion mentioned is located in a repetitive sequence. these can be difficult to sequencelassemble depending on the method used. Can the authors be sure this is not the result of a sequencing or assembly error?

We are sure is an actual deletion. The deletion is located in the central part of one of the fragment sequenced, the fragment 23 (following Hassanin et al. 2009). Since we sequenced both strands using Sanger method for the 23 fragments, we are sure the delection is real. Besides the same deletion has been previously detected in other samples in our laboratory using different sets of primers (Rodriguez et al., 2010, Perez et al., 2014). Furthermore, the difference in band size due to the deletion could be seen in the agarose gels when amplified with the primers used in Rodriguez et al. (2010)

Fig. 2. What do the colored rectangles represent?

The colored rectangles highlight the subclades well supported, within the main clades. Explanation has been included in the figure caption.

the squares are printed by default by the figure programme and contain those nodes that reach a certain confidence value.

The supplement file only contained one extra tree and didn't have any legend, so I couldn't assess this file.

The three extra trees were sent in three different pdf files, probably there was some problem with the uploading that I missed. It has been corrected and now all the trees are in the same pdf file with their corresponding legend

There are multinle instances of Gen-Bank in the text Please correct

All the Gen-Bank and Genbank references have been changed to GenBank